# Visual odometry of *Rhinecanthus aculeatus* depends on the visual density of the environment

Cecilia Karlsson [1] ✉, Jay Willis [1], Matishalin Patel[1] & Theresa Burt de Perera [1] ✉

Distance travelled is a crucial metric that underpins an animal's ability to navigate in the short-range. While there is extensive research on how terrestrial animals measure travel distance, it is unknown how animals navigating in aquatic environments estimate this metric. A common method used by land animals is to measure optic flow, where the speed of self-induced visual motion is integrated over the course of a journey. Whether freely-swimming aquatic animals also measure distance relative to a visual frame of reference is unclear. Using the marine fish *Rhinecanthus aculeatus*, we show that teleost fish can use visual motion information to estimate distance travelled. However, the underlying mechanism differs fundamentally from previously studied terrestrial animals. Humans and terrestrial invertebrates measure the total angular motion of visual features for odometry, a mechanism which does not vary with visual density. In contrast, the visual odometer used by *Rhinecanthus acuelatus* is strongly dependent on the visual density of the environment. Odometry in fish may therefore be mediated by a movement detection mechanism akin to the system underlying the opto-motor response, a separate motion-detection mechanism used by both vertebrates and invertebrates for course and gaze stabilisation.

[1] Department of Zoology, Research and Administration Building, 11a Mansfield Road, Oxford OX1 3SZ, UK. ✉email: hclk2@cam.ac.uk; theresa.burt@zoo.ox.ac.uk

Path integration is a powerful mechanism that enables animals to find their way home efficiently. To achieve this, individuals must be able to determine the direction and distance of their travel trajectories[1]. To obtain directional information, many terrestrial animals, both vertebrate and invertebrate, possess diverse compasses based on the Earth's magnetic field[2,3], overhead polarised light[4,5], the direction of sunlight[5] or the moon and stars[6], and proprioceptive inputs from body rotations[1]. In order to discern travel distances, different species of terrestrial animals rely on self-induced optic flow[7–11], stride integration[12–15], energy use[16] and internal vestibular cues[17] in combination or in isolation.

While a concerted interest has been directed towards terrestrial species, virtually nothing is known of how freely-swimming aquatic animals such as the large and early-branching vertebrate group, the teleost fish, use self-motion information to navigate by path integration. This paucity of information has fundamental implications for our understanding of how universal the mechanisms of navigation are, and crucially what shapes the evolution of these mechanisms.

Here, we explore for the first time how a pelagic fish measures distance travelled. One common mechanism through which many land animals—including humans[11,18] and invertebrates such as the honeybee[7,10,19], desert ant[9] and wolf spider[8]—measure distance travelled is using self-induced optic flow. Self-induced optic flow broadly refers to the speed at which visual information is perceived to pass by an animal as a result of its own movement. This information, if integrated over time, can provide a reliable indication of distance travelled.

Animals navigating in water also have the potential to detect and use information relative to a fixed visual background. However, they would do so against the constraints of rapidly attenuating light with depth and distance, and with access to additional sensory information from the moving fluid medium through which they swim. It is therefore unclear if fish would have evolved to measure distance travelled in a similar way to land animals, or whether the constraints of navigating through water have resulted in use of specific sensory mechanisms adapted for reliable and robust information acquisition in water. We address this question by testing whether a shallow water marine fish, *Rhinecanthus aculeatus*, relies on optic flow information for odometry, and explore how this visual information is processed. *Rhinecantus aculeatus* is an intertidal marine fish, navigating between its home nest and foraging locations over undifferentiated coral rubble. In such an environment, prominent landmarks are scarce and temporally unreliable. Navigating using distance and direction information derived from self-movement is therefore likely to be central to their survival and success.

Wild-caught *Rhinecanthus aculeatus* subjects were trained to swim a distance of 0.80 m under an overhead infrared proximity sensor controlling aquarium lights running along the top of a linear tunnel (Fig. 1a). Swimming beneath this detector switched on the surrounding aquarium lights and signalled to the trained fish to return to the start area for a food reward. Lateral and ventral walls of the tunnel were patterned with alternating black and white stripes of width 0.02 m, oriented perpendicular to the direction of movement (Fig. 1, test 1). The fish were then tested in their ability to reproduce this learned distance when the infrared detector controlling the overhead lights was removed, first under the training visual condition (Fig. 1b, test 1) and again following manipulations to the visual background (Fig. 1b, test 2–4). Visual manipulations either altered the translational spatial frequency of the visual pattern (test 2), the geometry of the visual pattern whilst keeping translational spatial frequency at the level experienced during training (test 3), or removed translational optic flow information (test 4). To ensure fish were only able to

use self-motion cues and not any absolute positional cues internal or external to the tank, the start area was shifted between three increasingly distal positions within the tunnel at the start of each training and testing session.

## Results and discussion

Figure 2 shows that odometry in *Rhinecanthus aculeatus* is visually driven and scene dependent. Manipulating the visual background produced consistent shifts in distance estimate distributions across all tested individuals. Under the training condition, the fish reproduced the learned distance with a population average underestimate of 0.0407 m (Fig. 2a, test 1). Distance travelled was calculated using self-motion information alone—we observed predictable systematic shifts in the absolute turning position in the tunnel according to start area position (Fig. 3, test 1. $\chi^2_2 = 109$, $p < 0.001$. Tukey pairwise comparisons: +0:+10—$z = 4.56$, $p < 0.001$; +10:+20—$z = 5.86$, $p < 0.001$; +0:+20—$z = 10.3$, $p < 0.001$). Fish were therefore not using any positional landmark cues external to the maze; were not generalising across start area positions; and, were not using looming stimuli internal to the maze such as learning the angle subtended by the wall to the floor at the end of the tunnel.

Removing translational optic flow by presenting horizontal stripes on the lateral and ventral walls prevented fish from being able to accurately reproduce the learned distance (Fig. 2, test 4). The average distance of 0.935 m was a significant overestimate compared to the baseline ($\chi^2_3 = 489$, $p < 0.001$; Tukey pairwise comparison Test 1:Test 4, $z = 5.26$, $p < 0.001$). One out of the five tested fish in this condition swam to the end of the tunnel on almost all trials (Fish D), but on average distance estimates had a significantly higher variance compared to the three treatments where fish had access to translational optic flow information ($\chi^2_3 = 275$, $p < 0.001$; Tukey pairwise comparisons—Test 1:Test 4, $z = 11.8$, $p < 0.001$; Test 2:Test 4, $z = 11.0$, $p < 0.001$; Test 3:Test 4, $z = 9.35$, $p < 0.001$).

In test 4, there was also no significant systematic shift in absolute turning position in the tunnel according to start area position. A significant pairwise difference for this treatment was only observed between start area positions +0:+20, and +0:+10 (whole model—$\chi^2_2 = 7.62$, $p = 0.0221$; Tukey pairwise comparisons +0:+20, $z = 2.43$, $p = 0.0402$; +0:+10, $z = 2.50$, $p = 0.0337$) and this was likely to have been driven by random individual responses to the removal of optic flow. Two fish (Fish D and E) swam to the end of the tunnel on some but not all trials. Trials in which fish swam to the end of the tunnel coincided with the start area in positions +10 and +20 for Fish D and +20 for Fish E. The three remaining fish exhibited no consistent pairwise shifts in turning position according to start area position. We therefore conclude that despite the apparent shifting trend for the summarised data in Fig. 3 test 4, fish were not able to accurately resolve distance travelled when translational optic flow information was removed.

Removing optic flow information also compromised swimming speed control, but to a much lesser extent than the ability to reproduce travel distances. Average swimming speed increased compared to the training condition by 14% ($\chi^2_3 = 102$, $p < 0.001$; Tukey pairwise comparison, Test 1:Test 4, $z = 3.58$, $p = 0.002$). This is in contrast to the three-fold increase in flight speed exhibited by honeybees flying down a tunnel with axial stripes[20]. However, the variance in swimming speed across distance estimates was consistently higher compared to all treatments where translational optic flow information was provided, suggesting visual motion information does have a small role in maintaining consistent swimming speeds (Fig. 4, $\chi^2_3 = 137$, $p < 0.001$; Tukey pairwise comparisons—Test 1:Test 4, $z = 8.49$, $p < 0.001$; Test 2:Test 4, $z = 8.32$, $p < 0.001$; Test 3:Test 4, $z = 7.50$, $p < 0.001$).

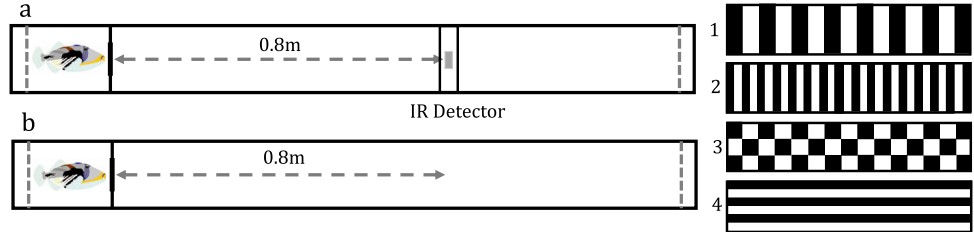

**Fig. 1 Testing the role of optic flow in distance estimation: training and testing set-up. a** Training—the fish was trained to swim 0.8 m to an overhead infrared proximity sensor (IR Detector) which when the fish passed beneath it, detected a voltage change and via the Arduino computer caused the aquarium lights to switch on, signalling to the fish to return to the start area for a food reward. Training was conducted with the tunnel walls and floor patterned with alternating 0.02-m-wide black and white vertical stripes (set-up 1). **b** Testing—during testing trials, the infrared proximity sensor controlling the overhead lights was removed. The fish was therefore unable to cause the lights to turn on during testing trials. This was to test whether the fish had learned the correct distance, or to swim to the infrared detector landmark to encounter the light stimulus. The fish was tested with four different background combinations: (1) 0.02 m vertical stripes; (2) 0.01 m vertical stripes; (3) 0.02 m wide checkerboard pattern; (4) horizontal stripes of width 0.02 m.

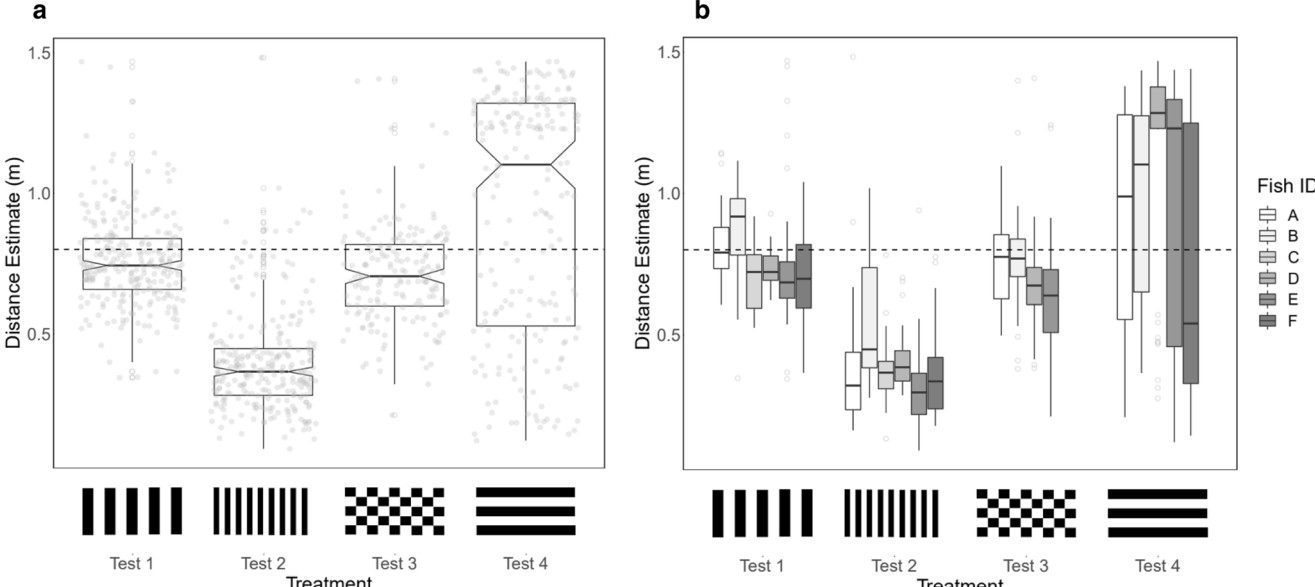

**Fig. 2 The impact of background visual patterns on estimates of distance made by _Rhinecanthus aculeatus_. a** Overall distance estimates across visual treatments. **b** Distance estimates across visual treatments, split across fish identity. Fish were trained to a target distance of 0.80 m (horizontal dashed line), and tested across four visual backgrounds. Population averages across treatments were as follows: test 1 mean = 0.759 m; test 2 mean = 0.397 m; test 3 mean = 0.710 m; test 4 mean = 0.935 m. Test 2 produced significant underestimates of distance travelled compared to all visual treatments, and test 4 produced significant overestimates compared to all visual treatments. Box plots indicate the median, interquartile range, and notches compare medians between groups. Grey points indicate individual distance estimates in figure (**a**) and outliers in figure (**b**). Sample sizes across fish and treatments were as follows. Test 1: n = 268 (By fish: A, n = 43; B, n = 45; C, n = 45; D, n = 45; E, n = 45; F, n = 45), Test 2: n = 265 (By fish: A, n = 44; B, n = 45; C, n = 44; D, n = 44; E, n = 43; F, n = 45), Test 3: n = 176 (By fish: A, n = 45; B, n = 44; D, n = 43; E, n = 44), Test 4: n = 216 (By fish: A, n = 45; B, n = 44; D, n = 44; E, n = 41; F, n = 42).

When using optic flow for odometry, movement speeds, and centreing responses through gaps and tunnels, terrestrial animals measure the angular motion of features across the retina, a mechanism that is independent of the spatial frequency of the environment[10,11,19,21–23]. However, Fig. 2 indicates that the fish visual odometer is highly sensitive to the spatial frequency of visual information. Doubling the spatial frequency (test 2) resulted in a large underestimation of travel distances compared to the training treatment (test 2 mean distance estimate = 0.397 m, Tukey pairwise comparison Test 1:Test 2—$z = -18.3$, $p < 0.001$). Any generalised novelty responses were quantified by testing fish with a checkerboard pattern of equivalent spatial frequency to the training condition (test 3). In test 3, distance estimates were restored close to the level of the training condition. A population mean of 0.710 m was recorded, a small

underestimation compared to the training condition (Tukey pairwise comparison Test 1:Test 3—$z = -3.08$, $p = 0.0110$). Despite this, there was a strong significant difference between test 2 and test 3 ($z = 14.4$, $p < 0.001$). We can therefore conclude that the large underestimates observed in test 2, with distance estimates reduced by approximately half, were responses to the doubling of spatial frequency and not a generalised novelty response.

In contrast, swimming speeds were only partially affected by the spatial frequency of the visual background. Under the training condition (test 1), fish swam at an average speed of 0.297 m/s. However, doubling the spatial frequency in test 2 resulted in only a small significant reduction in swimming speed compared to both tests 1 and 3 (Fig. 4a). Swimming speeds on average declined by 19% compared to test 1 and 13% compared to test 3 (test 2

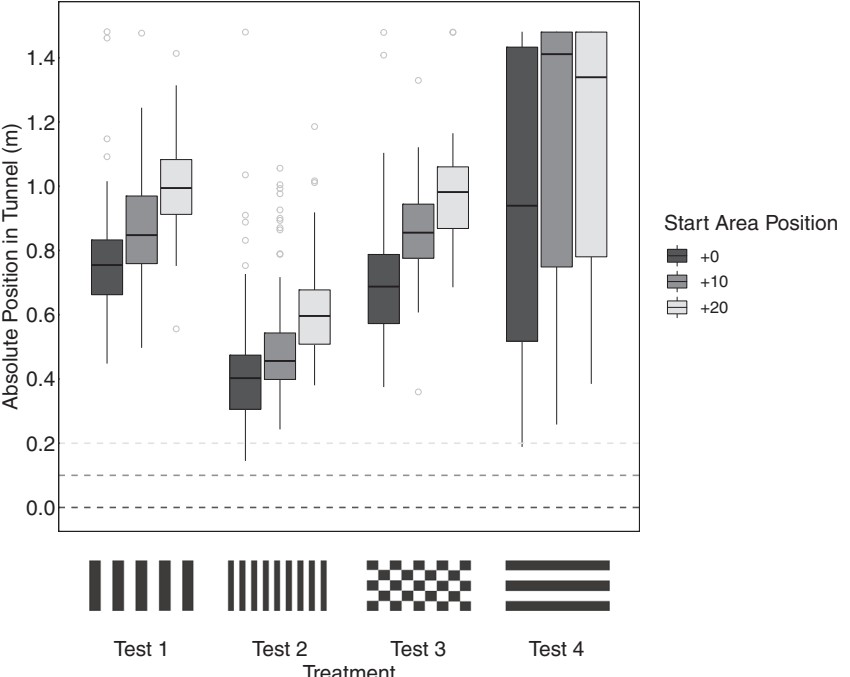

**Fig. 3 Absolute turning positions across treatments.** Between sessions, the start area was moved across three increasingly distal positions within the tunnel to control for the use of external cues and assess distance estimation abilities. If fish were using an internal measure of distance, there would be a systematic shift in turning position according to each start area position. This was indeed observed across treatments 1–3 where spatial frequency information was provided, indicating that fish are reporting perceived travel distance. When spatial frequency information was removed in treatment 4, there was no consistent significant shift in turning position for all start area positions. Grey dashed lines indicate corresponding start area positions for the three start area positions (dark grey: +0; mid grey: +10; light grey: +20). Box plots indicate the median, interquartile range, and grey points indicate outliers. Each fish completed a maximum of 15 distance estimates for each start area position.

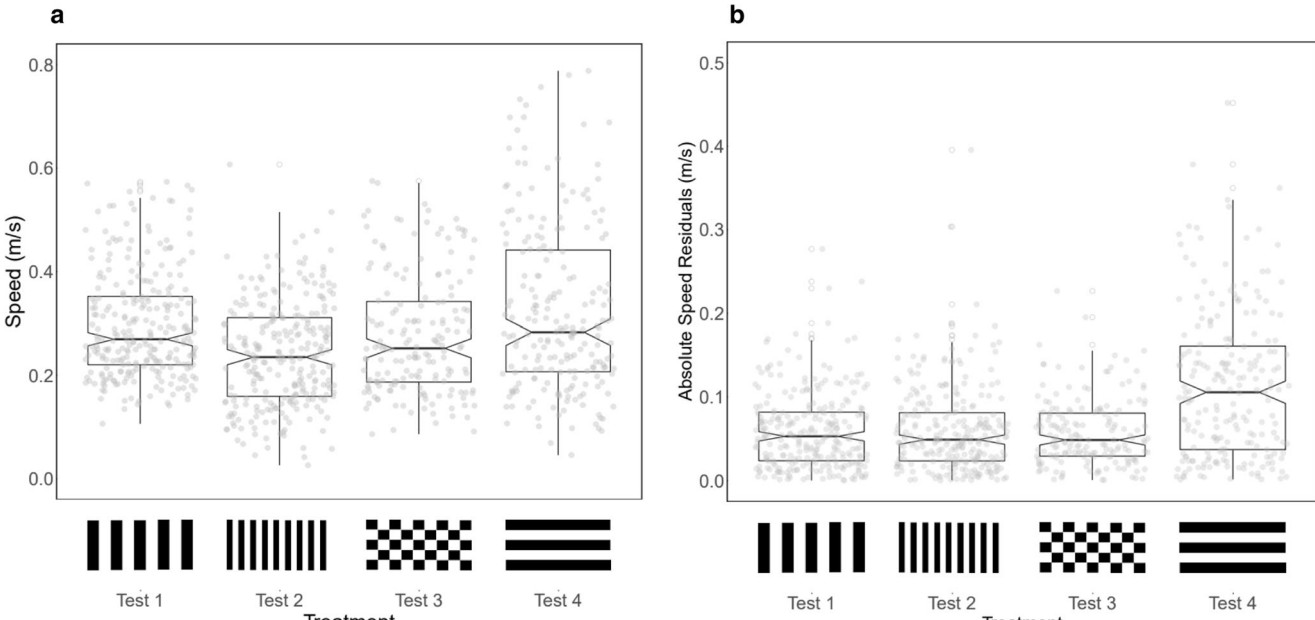

**Fig. 4 Swimming speeds across treatments. a** Mean swimming speeds varied with visual treatment but not proportionately with the modifications in spatial frequency. **b** The variability in swimming speed, measured as the residual swimming speeds from the treatment mean, increased with removal of translational optic flow. Box plots indicate the median, interquartile range, and notches compare medians between groups. Grey points indicate individual trials. Sample sizes across fish and treatments were as follows. Test 1: $n = 268$ (By fish: A, $n = 43$; B, $n = 45$; C, $n = 45$; D, $n = 45$; E, $n = 45$; F, $n = 45$), Test 2: $n = 265$ (By fish: A, $n = 44$; B, $n = 45$; C, $n = 44$; D, $n = 44$; E, $n = 43$; F, $n = 45$), Test 3: $n = 176$ (By fish: A, $n = 45$; B, $n = 44$; D, $n = 43$; E, $n = 44$), Test 4: $n = 216$ (By fish: A, $n = 45$; B, $n = 44$; D, $n = 44$; E, $n = 41$; F, $n = 42$).

mean speed = 0.240 m/s; test 3 mean speed = 0.276 m/s. Tukey pairwise comparison, Test 1:Test 2—$z = -6.42$, $p < 0.001$; Tukey pairwise comparison, Test 2:Test 3—$z = 2.78$, $p = 0.0277$). Therefore, while the relative change in swimming speed across visual treatments is the same as that observed for distance estimates, the absolute magnitude of this change is much lower. This suggests that visual inputs do play a role in maintaining consistent, target swimming speeds using the same motion-sensitive pathways underpinning odometry, but that this behaviour is also —and perhaps predominantly—controlled by other sensory mechanisms.

Overall, we have shown that teleost fish do use visual motion information to estimate distance travelled. However, the underlying mechanism differs fundamentally from previously studied terrestrial animals such as humans and honeybees. In order to estimate travel distances, terrestrial animals measure the rate of angular movement of visual features over the retina (sometimes referred to as 'true optic flow') and integrate this information over time[10,11,21,23]. This mechanism depends strongly on the distance the moving animal is from a visual background as well as its movement speed. For example, a bee passing close to a visual feature will experience a faster rate of angular change than a bee moving at the same speed but further away from the visual feature. In practice, bees use the same visual motion mechanism for odometry and flight speed control. Bees moving through cluttered environments will therefore fly at a lower speed compared to when flying through open environments[21,24]. By doing this, they ensure they experience the same rate of optic flow throughout any journey irrespective of the depth and texture of the visual scene[10,21,24]. Crucially, this mechanism is independent of the spatial frequency of the environment[10,19,22,23].

In contrast, the distance estimates produced by *Rhinecanthus aculeatus* in our tunnel experiment vary strongly with the spatial frequency of the visual background. Distance estimates are related to the number of contrast changes experienced en-route to the point of interest during training. Teleost fish do not therefore appear to measure the angular rate of movement of visual features to estimate travel distances. Instead, we suggest that *Rhinecanthus'* visual odometer is mediated by a movement-detection mechanism similar to that underlying the optomotor response. The optomotor response is a reflex resulting in head and eye movements that stabilise the visual field, allowing animals to correct for any unwanted deviations such as displacements caused by water currents or gusts of wind[21,25,26]. This mechanism depends on the rate of movement of visual edges, or the spatio-temporal frequency, of the visual environment. For example, when an animal with an optomotor response is placed in a rotating drum patterned with vertical gratings, it will turn in the same direction as the movement of the visual scene, thereby stabilising its orientation relative to its surroundings[24,25]. In a similar way to the distance estimates reported in this study, this turning response is highly dependent on the spatial frequency of the rotating visual gratings, and no head or body rotations are observed in the absence of translational spatial frequency information[25].

In zebrafish, the optomotor response is mediated by direction-sensitive neurons in the pretectal area, directly ventral to the optic tectum[27]. These neurons are tuned to a restricted range of spatial frequencies and wavelengths, and are sensitive to variations in spatial contrast[25,27,28]. Whether the same neural architecture underlies sensory processing in the teleost visual odometer and the optomotor response remains to be seen. Future work quantifying the behavioural output of the teleost odometer in response to a range of visual scene changes will allow us to further compare the mechanisms underlying these two behaviours.

An odometry mechanism that measures the spatio-temporal frequency of the visual environment is less robust to visual variability. However, *Rhinecanthus* typically forages over areas of undifferentiated coral rubble and sand, an environment that is highly structured but regular. If a fish travels through scenes with regular spatial frequencies at a near constant speed, and integrates this visual information to return to their home nest through the same environment, the underlying movement-detecting neurons will show the same pattern of responses on the outward and return journeys. In this case, using a mechanism that depends on spatial texture is therefore not likely to compromise navigation accuracy. This mechanism does, however, rely on the fish maintaining constant swimming speeds.

Previous work with zebrafish has found no evidence of visual motion-mediated swimming speed control in teleost fish[29]. Teleosts are instead thought to regulate swimming speeds predominantly through hydrodynamic inputs to the lateral line[30]. However, we show that swimming speeds in *Rhinecanthus aculeatus* are at least partly controlled by the same spatial frequency-dependent visual pathway as for distance travelled. We suggest that this visual speed estimate is then integrated with sensory information from other non-visual sensory modalities to set the final swim speed of the animal. The result is that in the presence of additional non-visual cues, such as hydrodynamic inputs to the lateral line, altering the visual information had only small effects on the final swimming speed of the animal. The small increased reliance on visual information by *Rhinecanthus aculeatus* compared to zebrafish may be due to ecological differences. *Rhinecanthus'* coral-reef habitat is both highly structured and brightly-lit, which could make using visual information for navigation and swimming control a more reliable source of information than it is for zebrafish occupying waters where the range of visual information may be more limited. Moreover, water flows in *Rhinecanthus'* intertidal habitat are highly variable in strength, direction, and turbulence. Therefore, integrating hydrodynamic information with visual inputs would permit more robust swimming speed control. Future work will be required to fully explore the interaction between visual motion and hydrodynamic information for both odometry and speed control in this species.

To conclude, we have shown that *Rhinecanthus aculeatus* uses visual motion information for odometry and speed control. However, it achieves this using a different mechanism to that seen in terrestrial animals. Both odometry and swimming speeds were sensitive to the presence or absence of translational optic flow, but also its spatial frequency. Moreover, while odometry appeared to be fully reliant on visual information, swimming speeds were predominantly controlled by alternative non-visual sensory mechanisms. Together, our results reveal that although the underlying mechanisms differ, optic flow information can be as valuable for performing navigation calculations underwater as it is on land.

## Methods

**Approvals**. This work did not fall under any regulated procedures outlined by ASPA, but was approved by a local ethics committee prior to starting experiments (AWERB—Animal Welfare Ethical Review Body, University of Oxford. Project code: APA/1/5/ZOO/NASPA/Burt/PathIntegration).

**Behavioural apparatus**. A linear Perspex maze measuring 0.25 m high × 0.16 m wide × 1.80 m long was built within a flow-through tank connected to the home water system to maintain constant water parameters. The walls and floor were interchangeable across 4 visual patterns, created using non-toxic black vinyl adhered to white Perspex. A perforated white screen was placed at either end to create laminar water flow whilst blocking the visual stimuli provided by cues external to the tank. A moveable start area of dimensions 0.25 m high × 0.16 m wide × 0.30 m long could be placed in one of three start area positions, all 0.1 m apart. An infrared proximity sensor (SHARP 2Y0A21 proximity sensor) was placed at water level, 0.80 m from the start area doorway. This was attached to an Arduino microprocessor which, through a Matlab (Mathworks Inc.) program, controlled the aquarium lights (Interpret Triple LED Lighting System, 0.75 m) running along the

top of the lateral maze walls. The voltage of the infrared (IR) detector varied between 0 and 3.5 V depending on the strength of the reflection from objects which passed in front of it. We tested the response of the detector to objects in water and after this we set a threshold of 1.7 V. As the fish passed beneath the detector, a voltage change was registered and when this exceeded 1.7 V, the aquarium lights switched on. A Point Grey Grasshopper 3M camera (FLIR Machine Vision Cameras) was placed 1.1 m above the water level to record testing trials.

**Behavioural training.** Six wild-caught Picasso triggerfish, *Rhinecanthus aculeatus*, were trained to swim 0.80 m with a visual background of vertical black and white stripes of width 0.02 m to an overhead infrared detector to switch on surrounding aquarium lights and return home for a food reward. This created a learned association between active swimming of a certain distance and a food reward. Training sessions continued until 10 correct trials were complete or until 10 min had passed, whichever came first. To control for use of external positional landmark cues, at the start of each session the start area was moved between three positions, located increasingly distally through the tank by 0.10 m increments. The infrared proximity sensor was moved accordingly to maintain the correct distance of 0.80 m. Training was considered complete when the fish swam directly out to the light flash and back on 80 percent of trials within the 10-min session time limit, across three consecutive sessions.

**Testing.** Fish were tested on their ability to reproduce the learned distance following a series of manipulations to their visual background. Testing conditions were as follows. Test 1 (training condition): alternating black and white vertical stripes of width 0.02 m. Test 2: Alternating black and white vertical stripes of width 0.01 m, doubling the translational spatial frequency. Test 3: A checkerboard pattern of width 0.02 m, restoring the translational spatial frequency to the training condition in order to test whether any distance estimate shifts in test 2 and 4 are novelty responses. Test 4: Alternating black and white horizontal stripes of width 0.02 m, removing any translational optic flow. The ordering of test treatments varied between fish. Testing sessions consisted of a 5 training + maximum 5 testing trial structure. The 5 training trials were conducted with a visual background of 0.02 m vertical black and white stripes, and fish were rewarded if they swam out directly to the infrared detector to turn on the lights and returned home with no additional turns in the tunnel. The purpose of this block was to ensure fish were at testing criterion (80% performance) prior to each set of testing trials. If performance dropped below 80%, fish were returned to training until they achieved testing criterion once more. If fish failed to reach testing criterion again even after 10 additional training sessions, they were removed from further testing. This was to account for any reduction in motivation between visual treatment testing blocks. All fish were motivated to complete testing sessions for test treatments 1 and 2, but a loss of motivation and training performance resulted in one fish being removed before treatment 4 (horizontal stripes), and two fish being removed from treatment 3 (checkerboard pattern). If the preceding 5 training trials were successful, the fish was subsequently returned to its home tank, the visual background manipulated, and then returned to the start area in the experiment tank and permitted to complete a maximum of 5 testing trials. Test trial blocks were of variable length according to individual differences and daily fluctuations in motivation, but up to a maximum of 5 consecutive trials. During these testing trials the infrared detector was removed, preventing the fish from turning on the surrounding aquarium lights at any point in the tunnel. The fish was released from the start area into the experiment arena and food rewards were randomly provided upon return to the start area to prevent frustration and extinction of the learned behaviour to return to the start area between trials. A successful testing trial, or distance estimate, was considered the maximum position of the fish's nose prior to turning back to the start area. On some testing trials, a food reward was required to tempt the fish back to the start area doorway, but this was only presented once the distance estimate was complete (the fish had turned 180°, or reached the end of the tunnel). A total of 15 testing trials were completed at each start area position for all four treatments, producing 45 estimates per treatment for each fish.

**Video analysis—extracting distance and speed data.** Testing sessions were recorded using an overhead camera (Point Grey Grasshopper 3M) at 50 frames per second and saved as Audio Video Interleave (avi) files using the Streampix 7 video capture software (Image Width: 2448 pixels; Image Height: 350 pixels). A distance estimate was considered the maximum distance of the fish's nose from a position level with the start area doorway prior to turning home. The fish was not required to fully return into the start area between testing trials, but was required to return level with the start area doorway to ensure distance estimates were always measured from the same point. This accounted for any loss of distance reporting abilities associated with the testing treatments. In some testing sessions, fish displayed stereotyped swimming behaviours indicative of disengagement with the behavioural task, and these trials were excluded from analysis if they followed a set of regular engaged trials within the same testing session. This accounted for any reduction in motivation across the duration of a testing session. If a testing session began with this stereotyped disengaged behaviour, these were included in analysis as long as the fish returned level with the start area doorway at the end of each trial; such trials were deemed to be representative of the reaction to the testing condition.

This was observed at a higher frequency in test 4 (removal of optic flow) compared to the other three treatments. Each successful testing trial was extracted into a series of jpeg images using the Streampix 7 program, making them compatible for analysis in a Matlab video tracking program. Fish were required to have a minimum of 40 successful trials per testing treatment to be included in analysis (see supplementary information for a summary of successful trial numbers by fish across treatments). The Matlab program tracked the pixel coordinate position of the fish's nose upon exiting the start area and the maximum point prior to turning home. The video frame number was also recorded for each of these two points. Using the program R (The R Project, version 3.6.1) the total pixel distance travelled was calculated as the difference between the exit position and turning position, and estimates were converted to metric distances using the following conversion: 14.4 pixels = 0.01 m. Swimming speed was calculated by dividing distance travelled by time taken in seconds, calculated by converting the time taken in frames to seconds using the video frame rate information.

## Statistics and reproducibility

*Distance estimates.* General mixed effects modelling was used to analyse the effects of treatment on distance estimates. The continuous skewed data was best fitted using a gamma distribution, with the subsequent model meeting assumptions of linearity and normality of residuals. An initial model was constructed with start area position and visual treatment as fixed effects, and fish identity as a random effect. This was corrected to the most parsimonious model removing the non-significant effect of start area position. Tukey pairwise comparisons were then used to perform multiple pairwise comparisons between treatments. Sample sizes across fish and treatments were as follows. Test 1: $n = 268$ (By fish: A, $n = 43$; B, $n = 45$; C, $n = 45$; D, $n = 45$; E, $n = 45$; F, $n = 45$), Test 2: $n = 265$ (By fish: A, $n = 44$; B, $n = 45$; C, $n = 44$; D, $n = 44$; E, $n = 43$; F, $n = 45$), Test 3: $n = 176$ (By fish: A, $n = 45$; B, $n = 44$; D, $n = 43$; E, $n = 44$), Test 4: $n = 216$ (By fish: A, $n = 45$; B, $n = 44$; D, $n = 44$; E, $n = 41$; F, $n = 42$).

*Start area position.* General mixed effects modelling was used to analyse the effect of start area position on absolute turning position in the tunnel, split by treatment. The data was once more fitted with a gamma distribution, and models were constructed with start area position as a fixed effect, and fish identity as a random effect. Tukey pairwise comparisons were then used to perform multiple comparisons across start area positions for each treatment. Each fish completed a maximum of 15 distance estimates for each start area position.

*Swimming speed.* General mixed effects models were constructed for the effect of visual treatment on average swimming speed and swimming speed residuals. Visual treatment was assigned as a fixed effect and fish identity as a random effect. The data for both models were best fitted using a gamma distribution, with models meeting assumptions of linearity and normality of residuals. Tukey comparisons subsequently performed pairwise comparisons between visual treatments. Sample sizes across fish and treatments were as follows. Test 1: $n = 268$ (By fish: A, $n = 43$; B, $n = 45$; C, $n = 45$; D, $n = 45$; E, $n = 45$; F, $n = 45$), Test 2: $n = 265$ (By fish: A, $n = 44$; B, $n = 45$; C, $n = 44$; D, $n = 44$; E, $n = 43$; F, $n = 45$), Test 3: $n = 176$ (By fish: A, $n = 45$; B, $n = 44$; D, $n = 43$; E, $n = 44$), Test 4: $n = 216$ (By fish: A, $n = 45$; B, $n = 44$; D, $n = 44$; E, $n = 41$; F, $n = 42$).

**Reporting summary.** Further information on research design is available in the Nature Research Reporting Summary linked to this article.

## Data availability

All data and R code used for the analysis of the data is provided in the supplementary information (Supplementary Data 1–3).

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

## Acknowledgements

We thank Tim Guildford, Ollie Padget, and members of the OxNav group for help with experimental design and analysis. We also thank three anonymous reviewers for feedback that improved the manuscript considerably. This work was supported in full by a Biotechnology and Biological Sciences Research Council grant (BB/M011224/1) and a St John's College Lamb and Flag studentship awarded to Cecilia Karlsson.

## Author contributions

C.K. conceived the project, performed the experiments and analysis, and wrote the paper. J.K.W. aided with experiment design and construction, and provided feedback on the manuscript. M.P. aided with project design, statistical analysis and figures. T.B.dP. oversaw the project, and provided feedback on the manuscript.

## Competing interests

The authors declare no competing interests.
