## [Peer Review File · Communications Biology]

Reviewers' comments:

Reviewer #1 (Remarks to the Author):

GENERAL

This interesting study investigates vision-based odometry in *Rhinecanthus aculeatus* and finds that, in this species, odometry is visually driven and scene-dependent. This is unlike the case in flying insects (at least, honeybees), where the odometer is again vision-based, but is independent of the visual texture of the scene (i.e. independent of its spatial frequency content). It is worth noting, though, that insects seem to possess at least two different types of visually driven motion-sensitive pathways (see, for example, [1]). The odometry pathway and the speed control pathway appear to record image velocity veridically, largely independently of the visual texture (spatial frequency). On the other hand, the insect's optomotor response (which is used for course stabilization) is strongly dependent on the spatial frequency of the grating stimulus (or temporal frequency, for a given rotational speed), and on the contrast of the grating. One possibility, then, is that visual odometry in the fish is mediated by a movement-detecting system that is akin to the system that mediates the optomotor response of insects. To examine this more closely, it would be useful to additionally investigate the influence of scene contrast. In insects, visual odometry and speed control are robust to variations in contrast - but the optomotor response is strongly dependent on contrast, increasing approximately as the square of the contrast at low-to-medium contrasts.

Figs. 2 and 3 demonstrate convincingly that visual odometry depends on the spatial frequency of the scene (or, equivalently, its temporal frequency, or edge encounter rate, for a given swimming speed). However, I am not sure about the validity of the conclusion that the control of swimming speed is dependent on a movement detection system that is also sensitive to spatial frequency (or edge encounter rate). Firstly, the arguments are rather unclear and confusing. Are the authors saying that swimming speed is regulated by a system that measures speed using cues based on the image angular velocity ('true' optic flow) as well as the temporal frequency (edge encounter rate) - and that the observed swimming speed is the result of a compromise between the two measures? To me, Fig. 4 suggests that the swimming speed is essentially independent of scene texture - the variations in mean speed across the four stimulus conditions are very small, even though they may be statistically significant. The finding that the speed is largely independent of visual texture suggests that the speed of swimming may be regulated by a different sensory modality, for example, one that senses the aquatic flow rate using the lateral line organs. There is substantial evidence to support this (see, for example, [2,3]).

As I see it, the clearest, most parsimonious, and least controversial interpretation of the data from this study would be that (a) odometry in *Rhinecanthus* is visually mediated, as is the case in flying insects, but, unlike insects, the odometry is not robust to variations in scene texture; and (b) unlike insects, locomotion speed is not controlled by sensing image movement, but probably by a different modality that senses the motion of the fish relative to the water. This hypothesis could be tested in further experiments by examining the control of swimming speed in the presence of a water current that moves with or against the swim direction, as well as the presence of flanking visual textures. That would help determine whether swimming speed is regulated by monitoring the image motion, or the motion relative to the water, or a combination of both types of sensory cues.

The finding that the odometry (estimation of distance traveled) depends on the spatial texture (or temporal frequency, or edge encounter rate) of the scene may not be detrimental if the fish uses the same navigational route every time, and travels at a more or less constant speed (which it seems to do), and experiences the same sequence of visual scenes along the way. The underlying movement-detecting neurons would then show the same pattern of responses (even though the response varies from location to location, depending upon the current scene texture) and the cumulative odometric reading would then always be the same, and it would not compromise the navigational ability of the fish. This may be worth pointing out in the Discussion.

The Discussion is somewhat lengthy and repetitive, and there are many statements in the

Methods, Results and Discussion sections (detailed below) that are somewhat obscure. It would help to clarify these statements, and to make the Discussion shorter and more succinct.

SPECIFIC

In Figures 2, 3 and 4: It would be useful to include labels identifying the Test numbers (Test 1, Test 2, etc) under the visual patterns shown in the horizontal axis

There are a number of descriptions and statements in the Results and Discussion sections that are rather unclear and obscure. These are listed below:

Training: I find the description of shaping of the behavior during the training to be unclear, even after reading the extended description of the training in the Supplementary Information. As I understand it, the cueing light is turned on only during the training (to signal that the fish has traveled the desired distance, at which point it has to turn back and swim to the starting position). The cueing light is never turned on during the tests, correct? This needs to be spelled out clearly – the description is ambiguous.

Figure 4 legend: Not clear what you mean by "Fish *compensate* swimming speed according to the spatial frequency information provided..." Are you referring to the quality of *regulation* of swimming speed (the size of the error bars), or to the mean value of the measured speed (which differs by small amounts across the tests)?

Lines 138 – 139: Not clear what you mean by "the variance in swimming speed across distance estimates was greater compared to treatments where spatial frequency information was provided (fig 4, test 4)". Are you saying the variance in speed in Test 4 was greater than in the other tests?

Lines 139- 142: Not clear what you mean by " The reliance on spatial frequency information for swimming speed control as well as distance information indicates that fish are likely to perceive distance not as the sum of spatial contrast information, but as the speed of visual contrast information experienced over time (optic flow)." This statement seems to be self-contradictory. What does "sum of spatial contrast information" mean? Do you mean the number of cycles (bright/dark edges) traversed during the journey? Or something else?

Lines 157-159: Not clear what you mean by "This was accompanied by a significant compensating reduction in swimming speed compared to baseline (fig 4, test 2) Tukey pairwise comparison (0.02m stripes: 0.01m stripes): $z = 6.16$, $p < 0.001$." Why is Fig 4, test 2 regarded as a baseline? And what are you comparing for 0.02m stripes vs 0.01m stripes? Distance estimation, or swimming speed?

In lines 170-171 you talk about a final test treatment that used a novel checkerboard pattern. In what sense was this different from the original checkerboard pattern? Would be useful to specify this and include an illustration.

References

- [1] MV Srinivasan (2011) Honeybees as a model for the study of visually guided flight, navigation, and biologically inspired robotics. *Physiological Reviews* 91, 389-411.
- [2] JC Liao (2006) The role of the lateral line and vision on body kinematics and hydrodynamic preference of rainbow trout in turbulent flow. *J Exp Biol* 209, 4077-4090.
- [3] JC Liao (2007) A review of fish swimming mechanics and behaviour in altered flows. *Phil Trans R Soc B* 362, 1973-1993.

Reviewer #2 (Remarks to the Author):

In their study "Visual odometry in a freely-swimming fish" the authors present evidence that

Rhinecanthus fish use optic flow based odometry to determine the distance they travelled. This is a very well constructed study, with solid statistical analysis and novel results. I have a number of comments and suggestions on all sections of the manuscript, and will detail these below.

l.34-36 while I agree that nothing is known about visual odometry in teleost fish, there are some previous studies on the visual navigation / orientation behaviour of teleosts with respect to optic flow information (Scholtyssek et al 2014 Biol Lett. 10(5): 20140279), and on the neural circuits underlying some optic flow induced behaviours (i.e. Kubo et al 2014 Neuron 81(6), P1344-1359).

l.48-49 since the frame of reference from fluid movements is used as an argument here to distinguish the sensory environment of fish from that of other animals using optic flow based navigation, I wonder why the same argument would not apply to insects - which also obtain additional information from measuring air flow while flying?

l.101 and 106 and other instances: the authors should word these instances more precisely: the horizontal stripes do not provide any translational optic flow cues - which is not to say, especially when the fish did not swim on a perfect horizontal line translating parallel to the tunnel walls, that they did not provide any optic flow cues. Moreover, the horizontal stripe pattern also contained spatial frequency information - but not in terms of translational optic flow. This might seem nit-picky but in order to not confuse the readers, it is important to use the terminology in the most precise way possible.

l.110-120 the tense here switches to present tense while reporting results

l.121 "no consistent systematic shift" - I would not entirely agree. The median of the absolute position in the horizontal condition showed the same increase for the three start positions as for the other patterns - just that there was no SIGNIFICANT difference between all medians, if I understand correctly (due to the increased variance in the data). Moreover, the authors reported a significant difference between starting position 0 and 20 - thus clearly there was also an effect with this pattern (suggesting that fish did still measure the distance travelled - but greatly overestimated it). But it does not show, in my opinion, that they could not determine the distance travelled any more.

l.127-135 I find the results on swimming speed very intriguing, especially since previous studies (on zebrafish, see Scholtyssek et al) did not show such an effect. Thus, I wonder if the fish in this experiment also showed other changes in their swim control that would be related to those of insect flight control, such as changes in the average position in the tunnel ("centring response" with patterns inducing strong translational optic flow) and related to that, changes in the tortuosity of the their swim paths. It seems the authors should have the tracks of the fish swimming through the setup, and thus could provide answers (figures) to these questions, which would be of great interest to many visual neuroethologists (as this would be one of the few indications not only for visual odometry in fish, but visual control of position during swimming in general). I would therefore encourage the authors to do this analysis and present the additional figures.

l. 186-187 I would recommend the authors tone down this statement a little. Previous authors have already shown the importance of optic flow for underwater orientation (numerous previous studies on the optomotor response and optokinetic reflex of teleost fish, as well as a study on the responses of harbor seals to translational optic flow (Gläser et al 2014 Plos One))

l. 198-200 would it be possible that what the fish actually learned was for how long they need to swim in the setup before returning and getting their reward. Then, because the swim speed was determined by the different patterns, fish would swim further (higher speed, same time) with the horizontal patterns, and less far (lower swim speed) with the higher spatial frequency vertical stripes? If this was a possibility, I would encourage the authors to discuss whether it was likely or not.

Fig. 4 I was a bit puzzled by the fact that the pairwise tests show statistically significant

differences ($p < 0.05$) between Test 1 and all other conditions despite the strong overlap in the boxplots (particularly Test 1 and Test 3). I appreciate that despite the small number of individuals, a large number of trials per condition was collected so that the power of the statistical tests was probably rather high. I would suggest to help readers to understand this visually, by plotting the model estimates with the data figures - so it becomes clearer how refined these estimates are, and thus why there are significant differences reported despite the overlapping data ranges. Moreover, for Fig. 4 it would also be interesting for the reader to see the effects of individual fish, and thus plot the data in addition to the summary plots separated by fish identity.

A general comment on the presentation of the statistical results: I really appreciated the thorough statistical testing performed by the authors, which included tests for individual effects. However, I thought readability of the manuscript suffered somewhat from reporting the details of the statistical results in the figure legends. I would suggest to use a more visual summary of the statistical tests in the figures (i.e. asterisks) and then report the details of the tests in a supplementary table or in the methods section.

Reviewer #3 (Remarks to the Author):

This is a novel study investigating if and how fish are able to estimate the distance travelled. The main finding is that triggerfish use visual information for odometry and this is the first example of a marine animal that uses visual odometry. The text is clear and well written and the methods appear to be sound. In general, I think that the finding that the triggerfish use visual information to estimate the distance to a point of interest is supported by the data. I find that the interpretation of the data with respect to the mechanism that the fish are using, however, to be somewhat misguided and the explanation of the proposed mechanism to be unclear. I think that, to achieve its full potential, the study requires some more careful consideration of the results and clarification of the mechanism the authors are proposing. I outline my specific concerns and suggestions for changes below.

Major concern 1

Generally, I find some confusion in the way the text describing the difference between the strategy used by honeybees and the triggerfish strategy the authors are proposing. For example, in the abstract, it is stated that the 'Distance in measured by integrating the rate of movement of visual textures'. This description fits for the honeybee strategy, that is, integrating optic flow over the course of the journey (optic flow is commonly defined as the rate of movement of visual texture, although technically it is the pattern of motion experienced on the retina as an animal moves through the world). What the data actually show is that the distance estimate of the fish is related to the number of contrast changes they experience on the way to the point of interest, rather than anything else. The authors should be more clear and consistent throughout the text with the terminology as well as with their description of the mechanism that the fish might be using.

One example, Line 148: I am confused about the definition of 'optical edge rate', I think that the authors are referring to temporal frequency, that is the number of contrast changes passing a point in the eye per unit time. The optomotor response of most animals (including fish) is sensitive to temporal frequency and, as this is an established term in the literature, I suggest the authors use this instead of introducing a new term. I also suggest that the authors consider and cite relevant literature showing this phenomenon (for which there are examples from fish), for example Maaswinkel and Li 2003 'Spatio-temporal frequency characteristics of the optomotor response in zebrafish'.

I do not understand the interpretation given on line 165. How is the data inconsistent with a strategy of stripe counting? I think that this is exactly what the data show as the fish swim half the distance when the spatial frequency of the pattern is doubled. I do not understand the argument that the errors should vary if this were the case. What is the logical basis for assuming that the distance estimate error profiles correlate with the density of information?

The argument made on line 172 that the fish were not stripe counting is unfounded. The spatial frequency of the check pattern was the same as the stripes, therefore a method of counting the

number of contrast changes at a certain point in the visual field would certainly yield exactly the same result.

Line 198: How would the fish adjust their swimming speed? How would they know what swimming speed they have?

I also do not think that there is strong support for the statement in the abstract that the proposed mechanism for the fish is 'more robust to the constraints of navigating underwater'. I suspect that this statement has arisen from a misunderstanding of how the honeybee odometer works. This is particularly clear from the statement on Line 195: Where has this been argued? The optic flow-based odometer proposed for honeybees does depend on the distance to the nearby obstacles (see Esch and Burns 1995 and Srinivasan et al. 1997), so it will not produce the same readout for flight in clutter as for flight in open spaces.

Major concern 2

My other major concern with the interpretation of the results relates to the speed data. The difference in speed recorded between the two different spatial frequencies may have yielded a statistically significant result but I am not at all convinced by the similarity of the box plots in Figure 4 that this difference is significant for the fish. The scale on the figure is lacking detail so it is difficult to tell the difference between the medians, but my guess is that it is not more than 0.0005 m/s, that is 5 mm/s. This is an extremely small difference in speed and certainly does not suggest that the pattern has had a strong effect on the fish. Furthermore, if the rate of the stripes was important for speed control, the fish should have reduced their speed by half and this is clearly not the case. The support given for the idea that increases in spatial frequency are interpreted as increases in speed is weak. The sentence starting on line 148 cites paper 11 (Mossio et al. 2008) but in this paper, I find no reference of how spatial frequency affects speed perception (e.g. section 4.3.1 in the Mossio et al.). To the contrary, it finds that distance estimation in humans is not affected by the texture on the ground, unless I have really missed something. Also, I would argue that the evidence for the effect of spatial frequency on walking speed in humans provided in paper 26 (Ludwig et al. 2018) is not at all strong, with the largest effect of floor pattern equating to a change in walking speed of ~0.02 m/s (at speeds of 1.2-1.4 m/s, i.e. the change represents 1.6% of the speed) and therefore do not think that this provides sufficient support for this statement. I suggest that the authors revise this sentence and the citations they use to support it and reconsider their interpretation of the speed results as well as their interpretation of the mechanism underlying distance estimation in the trigger fish.

One previous study has reported the use of optic flow in fish for swimming control and this is relevant for the current study, especially the discussion on speed control: Scholtyssek C, Dacke M, Baird E* 2014 Control of self-motion in dynamic fluids, fish do it differently from bees. *Biol Lett* 10:20140279 doi: 10.1098/rsbl.2014.0279. For example, the finding that speed in the triggerfish increases with the horizontal stripes is contrary to the findings of Scholtyssek et al. and may suggest some variation in how fish ecology affects how they use visual information to control their swimming.

Comments/suggestions for changes

I suggest that the authors include a figure showing the raw trajectories of the fish. Do they swim in the centre of the tunnel, or closer to one wall? This has implications for how they perceive the spatial frequency of the patterns tested as this will vary with the viewing distance.

Throughout much of the text, symbols have been replaced with squares (e.g. in the legend for figure 4). This may just be a problem with the program but check to make sure.

Line 151: the sentence starting here seems out of place in the flow of the text, I suggest putting it before the preceding sentence.

Line 315: add an 's' to session

Line 316: '...at the start of each session' instead of '...each session start'

Line 341: Checkerboard does not need to be capitalised

Dear Editor,

We hereby submit a revision of our article "Visual odometry in a freely-swimming fish". We thank you and the reviewers for their positive and very careful comments, which have greatly improved our manuscript. We have dealt with all of the comments/suggestions and we hope that you will now find our manuscript suitable for publication. Below, we detail our responses to each individual point.

Yours sincerely,

Corresponding author
(on behalf of the co-authors)

Reviewer 1:

*Comment: This interesting study investigates vision-based odometry in *Rhinecanthus aculeatus* and finds that, in this species, odometry is visually driven and scene-dependent. This is unlike the case in flying insects (at least, honeybees), where the odometer is again vision-based, but is independent of the visual texture of the scene (i.e. independent of its spatial frequency content). It is worth noting, though, that insects seem to possess at least two different types of visually driven motion-sensitive pathways (see, for example, [1]). The odometry pathway and the speed control pathway appear to record image velocity veridically, largely independently of the visual texture (spatial frequency). On the other hand, the insect's optomotor response (which is used for course stabilization) is strongly dependent on the spatial frequency of the grating stimulus (or temporal frequency, for a given rotational speed), and on the contrast of the grating. One possibility, then, is that visual odometry in the fish is mediated by a movement-detecting system that is akin to the system that mediates the optomotor response of insects. To examine this more closely, it would be useful to additionally investigate the influence of scene contrast. In insects, visual odometry and speed control are robust to variations in contrast - but the optomotor response is strongly dependent on contrast, increasing approximately as the square of the contrast at low-to-medium contrasts.*

Response: We are grateful for this observation, and have amended the manuscript to include comments on the similarity between the visual odometer reported on in our experiment and the previously documented fish optomotor response. This is reflected in lines 19-23 of the summary, and lines 194-215 in the discussion of the main manuscript. We agree that it would be interesting to examine the effects of visual contrast on speed and odometry more closely, although this would constitute a slightly different question to those that we seek to answer in the current study, and would be part of a separate piece of work. We are unable to undertake this at present, but we have proposed it as an area for further study in lines 210-215.

Comment: Figs. 2 and 3 demonstrate convincingly that visual odometry depends on the spatial frequency of the scene (or, equivalently, its temporal frequency, or edge encounter rate, for a given swimming speed). However, I am not sure about the validity of the conclusion that the control of swimming speed is dependent on a movement detection system that is also sensitive to spatial frequency (or edge encounter rate). Firstly, the arguments are rather unclear and confusing. Are

the authors saying that swimming speed is regulated by a system that measures speed using cues based on the image angular velocity ('true' optic flow) as well as the temporal frequency (edge encounter rate) – and that the observed swimming speed is the result of a compromise between the two measures? To me, Fig. 4 suggests that the swimming speed is essentially independent of scene texture - the variations in mean speed across the four stimulus conditions are very small, even though they may be statistically significant. The finding that the speed is largely independent of visual texture suggests that the speed of swimming may be regulated by a different sensory modality, for example, one that senses the aquatic flow rate using the lateral line organs. There is substantial evidence to support this (see, for example, [2,3]).

Response: We agree with the reviewer that the swimming speed results required re-interpreting. We have now carried out an additional analysis, which suggests that swimming speeds are largely independent of spatial frequency, unlike the visual odometer. In lines 173-180 we quantify the changes in swimming speed according to spatial frequency, indicating only a 13% reduction in speed if spatial frequency is doubled. We have also extended the analysis of swimming speed to also look at swimming speed control by testing the variance of swimming speed as well as the mean swimming speeds (fig 4a and 4b respectively). While spatial frequency changes have only small effects on swimming speeds, removing optic flow entirely in test 4 produced a highly significant increase in mean swimming speed ($p < 0.001$) and an increase in swimming speed variability ($p < 0.001$). This is discussed in lines 138-144, where we suggest that the visual control of swimming speed and visual odometry could be underpinned by separate mechanisms and presents an interesting area for future study.

Comment: As I see it, the clearest, most parsimonious, and least controversial interpretation of the data from this study would be that (a) odometry in Rhinecanthus is visually mediated, as is the case in flying insects, but, unlike insects, the odometry is not robust to variations in scene texture; and (b) unlike insects, locomotion speed is not controlled by sensing image movement, but probably by a different modality that senses the motion of the fish relative to the water. This hypothesis could be tested in further experiments by examining the control of swimming speed in the presence of a water current that moves with or against the swim direction, as well as the presence of flanking visual textures. That would help determine whether swimming speed is regulated by monitoring the image motion, or the motion relative to the water, or a combination of both types of sensory cues.

Response: While we agree that odometry and swimming speeds seem to be underpinned by separate mechanisms, we argue that swimming speed in *Rhinecanthus* is still visually controlled (lines 138-144 of results, and lines 225-237 of discussion), and may reflect differences between fish occupying different visual ecologies. This is covered in our response to the previous comment, where we have included additional analysis on swimming speed variability as well as average speed.

Comment: The finding that the odometry (estimation of distance traveled) depends on the spatial texture (or temporal frequency, or edge encounter rate) of the scene may not be detrimental if the fish uses the same navigational route every time, and travels at a more or less constant speed (which it seems to do), and experiences the same sequence of visual scenes along the way. The

underlying movement-detecting neurons would then show the same pattern of responses (even though the response varies from location to location, depending upon the current scene texture) and the cumulative odometric reading would then always be the same, and it would not compromise the navigational ability of the fish. This may be worth pointing out in the Discussion.

Response: We have included this observation in lines 216-224 of the discussion.

Comment: The Discussion is somewhat lengthy and repetitive, and there are many statements in the Methods, Results and Discussion sections (detailed below) that are somewhat obscure. It would help to clarify these statements, and to make the Discussion shorter and more succinct.

Response: The discussion has undergone a full re-write in light of reviewer comments. We hope that this has dealt with the issues that have been highlighted here.

Comment: In Figures 2, 3 and 4: It would be useful to include labels identifying the Test numbers (Test 1, Test 2, etc) under the visual patterns shown in the horizontal axis

Response: We agree with this observation, and have done as suggested.

Comment: There are a number of descriptions and statements in the Results and Discussion sections that are rather unclear and obscure. These are listed below:

Training: I find the description of shaping of the behavior during the training to be unclear, even after reading the extended description of the training in the Supplementary Information. As I understand it, the cueing light is turned on only during the training (to signal that the fish has traveled the desired distance, at which point it has to turn back and swim to the starting position). The cueing light is never turned on during the tests, correct? This needs to be spelled out clearly – the description is ambiguous.

Response: We have clarified this in lines 71-84 of the main text as well as the text under the figure. The methods sections for both training and testing have also been extended with additional information to communicate more clearly how the behaviour was shaped and how testing sessions were carried out.

*Comment: Figure 4 legend: Not clear what you mean by "Fish *compensate* swimming speed according to the spatial frequency information provided..." Are you referring to the quality of *regulation* of swimming speed (the size of the error bars), or to the mean value of the measured speed (which differs by small amounts across the tests)?*

Response: The figure 4 legend has been updated according to a changed interpretation and analysis of the swimming speed results.

Comment: Lines 138 – 139: Not clear what you mean by "the variance in swimming speed across distance estimates was greater compared to treatments where spatial frequency information was

provided (fig 4, test 4)". Are you saying the variance in speed in Test 4 was greater than in the other tests?

Response: Yes, this is what we mean. We have included an additional figure (figure 4b) to illustrate this, and we have explained this result more explicitly in lines 138-144 to make this clearer as it forms a more central part of this paper than initially interpreted.

Comment: Lines 139- 142: Not clear what you mean by " The reliance on spatial frequency information for swimming speed control as well as distance information indicates that fish are likely to perceive distance not as the sum of spatial contrast information, but as the speed of visual contrast information experienced over time (optic flow)." This statement seems to be self-contradictory. What does "sum of spatial contrast information" mean? Do you mean the number of cycles (bright/dark edges) traversed during the journey? Or something else?

Response: This observation has been removed and the manuscript corrected in line with this comment.

Comment: Lines 157-159: Not clear what you mean by "This was accompanied by a significant compensating reduction in swimming speed compared to baseline (fig 4, test 2) Tukey pairwise comparison (0.02m stripes: 0.01m stripes): $z = 6.16, p < 0.001$." Why is Fig 4, test 2 regarded as a baseline? And what are you comparing for 0.02m stripes vs 0.01m stripes? Distance estimation, or swimming speed?

Response: We have more explicitly compared and quantified swimming speeds in lines 173-180, and separated presentation of swimming speeds and distance estimate results more clearly.

Comment: In lines 170-171 you talk about a final test treatment that used a novel checkerboard pattern. In what sense was this different from the original checkerboard pattern? Would be useful to specify this and include an illustration.

This is a misunderstanding due to unclear writing on our behalf. We were referring to the checkerboard pattern in test 3. We have corrected this to remove the word 'novel' from this sentence.

References

- [1] MV Srinivasan (2011) Honeybees as a model for the study of visually guided flight, navigation, and biologically inspired robotics. *Physiological Reviews* 91, 389-411.
- [2] JC Liao (2006) The role of the lateral line and vision on body kinematics and hydrodynamic preference of rainbow trout in turbulent flow. *J Exp Biol* 209, 4077-4090.
- [3] JC Liao (2007) A review of fish swimming mechanics and behaviour in altered flows. *Phil Trans R Soc B* 362, 1973-1993.

Reviewer 2:

In their study "Visual odometry in a freely-swimming fish" the authors present evidence that Rhinecanthus fish use optic flow based odometry to determine the distance they travelled. This is a very well-constructed study, with solid statistical analysis and novel results. I have a number of comments and suggestions on all sections of the manuscript, and will detail these below.

Comment: l.34-36 while I agree that nothing is known about visual odometry in teleost fish, there are some previous studies on the visual navigation / orientation behaviour of teleosts with respect to optic flow information (Scholtyssek et al 2014 Biol Lett. 10(5): 20140279), and on the neural circuits underlying some optic flow induced behaviours (i.e. Kubo et al 2014 Neuron 81(6), P1344-1359).

Response: We have been more explicit that this is the first time we are testing a visual odometer in teleost fish, not use of visual motion for navigation more generally. We also refer to the recommended studies in lines 170-171 and 188-190.

Comment: l.48-49 since the frame of reference from fluid movements is used as an argument here to distinguish the sensory environment of fish from that of other animals using optic flow based navigation, I wonder why the same argument would not apply to insects - which also obtain additional information from measuring air flow while flying?

Response: This is a very interesting point, which we agree is consistent with our study. However, we feel that this is beyond the remit of our manuscript, and we would not like to speculate on this without further work.

Comment: l.101 and 106 and other instances: the authors should word these instances more precisely: the horizontal stripes do not provide any translational optic flow cues - which is not to say, especially when the fish did not swim on a perfect horizontal line translating parallel to the tunnel walls, that they did not provide any optic flow cues. Moreover, the horizontal stripe pattern also contained spatial frequency information - but not in terms of translational optic flow. This might seem nit-picky but in order to not confuse the readers, it is important to use the terminology in the most precise way possible.

Response: We have updated this to be consistent and precise throughout the manuscript.

Comment: l.110-120 the tense here switches to present tense while reporting results

Response: We have corrected this to keep the use of past tense throughout.

Comment: l.121 "no consistent systematic shift" - I would not entirely agree. The median of the absolute position in the horizontal condition showed the same increase for the three start positions as for the other patterns - just that there was no SIGNIFICANT difference between all medians, if I understand correctly (due to the increased variance in the data). Moreover, the

authors reported a significant difference between starting position 0 and 20 - thus clearly there was also an effect with this pattern (suggesting that fish did still measure the distance travelled - but greatly overestimated it). But it does not show, in my opinion, that they could not determine the distance travelled any more.

Response: We think that this comment is a result of us not explaining our results fully enough. We have expanded the explanation of this result in the main body of the manuscript (lines 92-103) to deal with this. We have also included a breakdown of individual fish responses in the supplementary information to illustrate the effect of individual fish on the outcome of the overall model (the random effect of fish ID was included in all models, and was significant). For all treatments where optic flow information is provided (test 1-3), a trending systematic shift in turning position according to start area position is largely seen for all fish. However, for test 4 there is no systematic shift in turning position for Fish A, B and F. We suggest that the trend observed in figure 3 and the significant difference reported between positions 0 and 20 are skewed by the responses of Fish D and E. Fish E randomly swims to the end of the tunnel only for position 20, and Fish D chooses to swim to the end of the tunnel for positions 10 and 20. We suggest that these stochastic responses have a large effect on the outcome of the statistical analysis used. Together, we have interpreted this as the fish losing the ability to accurately estimate distance travelled by being unable to distinguish the appropriate turning positions according to start area position.

Comment: l.127-135 I find the results on swimming speed very intriguing, especially since previous studies (on zebrafish, see Scholtyssek et al) did not show such an effect. Thus, I wonder if the fish in this experiment also showed other changes in their swim control that would be related to those of insect flight control, such as changes in the average position in the tunnel ("centring response" with patterns inducing strong translational optic flow) and related to that, changes in the tortuosity of their swim paths. It seems the authors should have the tracks of the fish swimming through the setup, and thus could provide answers (figures) to these questions, which would be of great interest to many visual neuroethologists (as this would be one of the few indications not only for visual odometry in fish, but visual control of position during swimming in general). I would therefore encourage the authors to do this analysis and present the additional figures.

Response: We do have the fish tracks to hand, but we are reluctant to complete this additional analysis for two reasons: (i) We cannot account for fish having learned the necessity to swim centrally through the tunnel in this set-up as the doorway from the start area to the main experiment arena was centrally positioned, and the infrared detector to switch on the lights was also centrally positioned over the tunnel. Fish may therefore have learned to swim centrally through the tunnel in order to ensure they switched on the aquarium lights independently of any available visual information (ii) All distance estimates are different lengths both within and between treatments (ranging from <10cm to 1.3m), and these vary strongly with visual treatment provided. This means that using these tracks to explore centralisation responses would not constitute a fully controlled study to test this aspect of visual swimming control and is likely to lead to inaccurate results. However, in light of the results from this paper and my observations during training, we have completed a separate, controlled experiment exploring

the lateral positioning and swimming speeds of *Rhinecanthus* subjects according to the orientation and availability of visual information. Fish were trained to swim repeatedly through the length of the tunnel and then filmed following a series of visual manipulations (sensu studies by Srinivasan, Dacke, Schiffner and Altshuler on bees, budgerigars and hummingbirds). We are currently preparing this as a separate manuscript for imminent submission. We chose not to submit the data together, as we felt that it detracted from the central questions in this study; in our view, it is a separate question and a separate study.

Comment: l. 186-187 I would recommend the authors tone down this statement a little. Previous authors have already shown the importance of optic flow for underwater orientation (numerous previous studies on the optomotor response and optokinetic reflex of teleost fish, as well as a study on the responses of harbor seals to translational optic flow (Gläser et al 2014 Plos One))

Response: This correction has been made and the discussion fully re-written.

Comment: l. 198-200 would it be possible that what the fish actually learned was for how long they need to swim in the setup before returning and getting their reward. Then, because the swim speed was determined by the different patterns, fish would swim further (higher speed, same time) with the horizontal patterns, and less far (lower swim speed) with the higher spatial frequency vertical stripes? If this was a possibility, I would encourage the authors to discuss whether it was likely or not.

As our further analysis and re-interpretation of our swimming speed results indicate that the visual control of swimming speed and odometry appear to be controlled by different mechanisms, and at the very least are not equally sensitive to variations in spatial frequency, we conclude that the fish cannot be using elapsed travel time as a proxy for distance travelled. If this were the case, we would expect swimming speeds to halve when spatial frequency was doubled, resulting in the same elapsed travel time for half the travel distance compared to the training condition. We actually observed only a 13% reduction in swimming speed but almost a 50% reduction in distance travelled.

Comment: Fig. 4 I was a bit puzzled by the fact that the pairwise tests show statistically significant differences ($p < 0.05$) between Test 1 and all other conditions despite the strong overlap in the boxplots (particularly Test 1 and Test 3). I appreciate that despite the small number of individuals, a large number of trials per condition was collected so that the power of the statistical tests was probably rather high. I would suggest to help readers to understand this visually, by plotting the model estimates with the data figures - so it becomes clearer how refined these estimates are, and thus why there are significant differences reported despite the overlapping data ranges. Moreover, for Fig. 4 it would also be interesting for the reader to see the effects of individual fish, and thus plot the data in addition to the summary plots separated by fish identity.

Response: This is a result of the use of mixed effects modelling and repeated measures of individual fish, resulting in a large number of data points used in the pairwise comparisons. To help with clarity, we have added notches to the box plots (reflecting the 95% CI of the median). We have also included results from individual fish in the supplementary information.

Comment: A general comment on the presentation of the statistical results: I really appreciated the thorough statistical testing performed by the authors, which included tests for individual effects. However, I thought readability of the manuscript suffered somewhat from reporting the details of the statistical results in the figure legends. I would suggest to use a more visual summary of the statistical tests in the figures (i.e. asterisks) and then report the details of the tests in a supplementary table or in the methods section.

Response: We did try this, but are reluctant to replace statistical reporting with asterisks owing to the large number of relevant comparisons. Adding asterisks for significant pairwise comparisons cluttered the graph and distracted from the very clear trends as shown by the boxplots themselves. Pairwise comparisons are intentionally summarised at the end of the figure legend to help with readability as much as possible.

Reviewer 3:

Comment: This is a novel study investigating if and how fish are able to estimate the distance travelled. The main finding is that triggerfish use visual information for odometry and this is the first example of a marine animal that uses visual odometry. The text is clear and well written and the methods appear to be sound. In general, I think that the finding that the triggerfish use visual information to estimate the distance to a point of interest is supported by the data. I find that the interpretation of the data with respect to the mechanism that the fish are using, however, to be somewhat misguided and the explanation of the proposed mechanism to be unclear. I think that, to achieve its full potential, the study requires some more careful consideration of the results and clarification of the mechanism the authors are proposing. I outline my specific concerns and suggestions for changes below.

Response: We are very grateful for the detailed feedback given, and have made changes accordingly.

Comment: Major concern 1

Generally, I find some confusion in the way the text describing the difference between the strategy used by honeybees and the triggerfish strategy the authors are proposing. For example, in the abstract, it is stated that the 'Distance is measured by integrating the rate of movement of visual textures'. This description fits for the honeybee strategy, that is, integrating optic flow over the course of the journey (optic flow is commonly defined as the rate of movement of visual texture, although technically it is the pattern of motion experienced on the retina as an animal moves through the world). What the data actually show is that the distance estimate of the fish is related to the number of contrast changes they experience on the way to the point of interest, rather than anything else. The authors should be more clear and consistent throughout the text with the terminology as well as with their description of the mechanism that the fish might be using.

Response: Following very helpful comments from all three reviewers, our interpretation of the mechanism has been altered. This has been corrected in the abstract (lines 13-21) and we have

more clearly explained the difference between the mechanism used by our fish and bees in the discussion (lines 136-208).

Comment: One example, Line 148: I am confused about the definition of 'optical edge rate', I think that the authors are referring to temporal frequency, that is the number of contrast changes passing a point in the eye per unit time. The optomotor response of most animals (including fish) is sensitive to temporal frequency and, as this is an established term in the literature, I suggest the authors use this instead of introducing a new term. I also suggest that the authors consider and cite relevant literature showing this phenomenon (for which there are examples from fish), for example Maaswinkel and Li 2003 'Spatio-temporal frequency characteristics of the optomotor response in zebrafish'.

Response: We have updated changed from using optical edge rate, to referring to the spatial and temporal frequency. We have also compared the mechanism used for visual odometry with the optomotor response in the discussion (lines 153-177)

Comment: I do not understand the interpretation given on line 165. How is the data inconsistent with a strategy of stripe counting? I think that this is exactly what the data show as the fish swim half the distance when the spatial frequency of the pattern is doubled. I do not understand the argument that the errors should vary if this were the case. What is the logical basis for assuming that the distance estimate error profiles correlate with the density of information? The argument made on line 172 that the fish were not stripe counting is unfounded. The spatial frequency of the check pattern was the same as the stripes, therefore a method of counting the number of contrast changes at a certain point in the visual field would certainly yield exactly the same result.

Response: We have since removed this interpretation, and instead conclude (lines 154-156): 'the distance estimates produced by *Rhinecanthus aculeatus* in our tunnel experiment vary strongly with the spatial frequency of the visual background. Distance estimates are related to the number of contrast changes experienced en route to the point of interest during training.'

Comment: Line 198: How would the fish adjust their swimming speed? How would they know what swimming speed they have? I also do not think that there is strong support for the statement in the abstract that the proposed mechanism for the fish is 'more robust to the constraints of navigating underwater'. I suspect that this statement has arisen from a misunderstanding of how the honeybee odometer works. This is particularly clear from the statement on Line 195: Where has this been argued? The optic flow-based odometer proposed for honeybees does depend on the distance to the nearby obstacles (see Esch and Burns 1995 and Srinivasan et al. 1997), so it will not produce the same readout for flight in clutter as for flight in open spaces.

Response: We agree that our original description of the honeybee odometer was unclear. In our fully re-written discussion, we more clearly describe the mechanism used by honeybees and contrast this with the mechanism used by our fish. The abstract has been updated accordingly.

Comment: Major concern 2

My other major concern with the interpretation of the results relates to the speed data. The difference in speed recorded between the two different spatial frequencies may have yielded a statistically significant result but I am not at all convinced by the similarity of the box plots in Figure 4 that this difference is significant for the fish. The scale on the figure is lacking detail so it is difficult to tell the difference between the medians, but my guess is that it is not more than 0.0005 m/s, that is 5 mm/s. This is an extremely small difference in speed and certainly does not suggest that the pattern has had a strong effect on the fish. Furthermore, if the rate of the stripes was important for speed control, the fish should have reduced their speed by half and this is clearly not the case. The support given for the idea that increases in spatial frequency are interpreted as increases in speed is weak. The sentence starting on line 148 cites paper 11 (Mossio et al. 2008) but in this paper, I find no reference of how spatial frequency affects speed perception (e.g. section 4.3.1 in the Mossio et al.). To the contrary, it finds that distance estimation in humans is not affected by the texture on the ground, unless I have really missed something. Also, I would argue that the evidence for the effect of spatial frequency on walking speed in humans provided in paper 26 (Ludwig et al. 2018) is not at all strong, with the largest effect of floor pattern equating to a change in walking speed of ~0.02 m/s (at speeds of 1.2-1.4 m/s, i.e. the change represents 1.6% of the speed) and therefore do not think that this provides sufficient support for this statement. I suggest that the authors revise this sentence and the citations they use to support it and reconsider their interpretation of the speed results as well as their interpretation of the mechanism underlying distance estimation in the triggerfish.

Response: We thank the reviewer for spotting an error in our graphing! The scale on the y-axis was out by 100x. This has now been corrected. We have also quantified the % changes in swimming speeds in lines 126-137. We re-interpret our swimming speed results in light of this change being very small for the fish in lines 188-202.

Our analysis and interpretation of the swimming speed data has been updated in lines 104-110 and 126-137 of the results section. We have also included additional analysis of swimming speed variance in figure 4b as this greatly adds support to our new hypothesis that triggerfish do use visual motion to control swimming speeds, and this mechanism more closely resembles that used by honeybees and humans.

We also agree that we misinterpreted the results from the human literature and have updated our discussion accordingly.

Comment: One previous study has reported the use of optic flow in fish for swimming control and this is relevant for the current study, especially the discussion on speed control: Scholtyssek C, Dacke M, Baird E 2014 Control of self-motion in dynamic fluids, fish do it differently from bees. Biol Lett 10:20140279 doi: 10.1098/rsbl.2014.0279. For example, the finding that speed in the triggerfish increases with the horizontal stripes is contrary to the findings of Scholtyssek et al. and may suggest some variation in how fish ecology affects how they use visual information to control their swimming.*

Response: We have included this comparison in lines 188-202.

Comment: I suggest that the authors include a figure showing the raw trajectories of the fish. Do they swim in the centre of the tunnel, or closer to one wall? This has implications for how they perceive the spatial frequency of the patterns tested as this will vary with the viewing distance.

Response: This was also suggested by reviewer 2 and was discussed fully there.

Comment: Throughout much of the text, symbols have been replaced with squares (e.g. in the legend for figure 4). This may just be a problem with the program but check to make sure.

Response: I can't see this happening in my word or latex file – this could be a problem with the viewing program used?

Comment: Line 151: the sentence starting here seems out of place in the flow of the text, I suggest putting it before the preceding sentence.

Comment: Line 315: add an 's' to session

Done.

Comment: Line 316: '...at the start of each session' instead of '...each session start'

Done.

Comment: Line 341: Checkerboard does not need to be capitalised

Done.

Reviewers' comments:

Reviewer #1 (Remarks to the Author):

I thank the authors for addressing most of the comments in my earlier review.

However, I still find it difficult to be convinced from the data that optic flow plays a major role in regulating the swimming speed. The speed increases by a very small fraction (less than 10%, Fig. 4) when the tunnel is lined with axial stripes (which should almost completely eliminate the optic flow cues). This suggests that the role of optic flow in regulating swimming speed is, at most, very minor. In honeybees, on the other hand, the flight speed increases by a factor of about 3 when the checkerboard patterns lining the tunnel wall are replaced by axial stripes [see, for example, Fig. 2, Barron and Srinivasan (2006)]. To my mind, the results of the experiments investigating the control of swimming speed strongly suggest that the major regulator of swimming speed is a non-visual cue, possibly hydrodynamic, as I had suggested in my earlier review, and cited a number of studies that have already provided evidence for this. I realize that investigating the role of hydrodynamic cues would be a new effort that is probably beyond the scope of the present study, but I think that it would be important to at least discuss this possibility clearly in the Discussion, because that is what the data seems to suggest. The authors now raise this possibility briefly in the revised Discussion and cite a new reference (Scholtysse et al., 2014), but dismiss it immediately on the grounds of species differences. The comprehensive review article on the control of fish swimming (Liao 2007) that I had alluded to in my earlier review, which the authors cite in the first version of the manuscript, but not in the second version, clearly states:

"For the majority of fishes, the two most important sensory modalities for swimming are vision and the lateral line sense. The lateral line system detects flow velocity and acceleration via a series of mechanosensory hair cells that are distributed on or just under the skin along the head and body (Coombs et al. 1989). Studies have shown that fishes use both hydrodynamic cues (Dijkgraaf 1963; Montgomery et al. 1997; Engelmann et al. 2000; Coombs et al. 2001) and high-contrast visual cues (Ingle 1971) to swim steadily in uniform flow or in still water. There is a degree of functional redundancy between these two modalities such that blocking the lateral line does not alter the ability to swim in uniform flow if vision is kept intact (Dijkgraaf 1963). In a more complex hydrodynamic environment devoid of visual cues, blocking the lateral line interferes with the ability to swim in steady and unsteady flows. Specifically, fish swimming in the dark without a functional lateral line (i.e. lacking both superficial and canal neuromasts) entrain behind a cylinder less than fish with an intact lateral line (Montgomery et al. 2003)."

Thus, there is a wealth of published evidence to indicate that the lateral line organs play an important role in the control of swimming. It seems to me that the present study would convey a more truthful and meaningful message to the reader if it concluded that optic flow has virtually no influence on swimming speed (which must therefore be regulated by other cues).

There seems to be an inconsistency in the numbering of the tests. In the main text, Tests 1, 2, 3 and 4 pertain to the broad vertical stripes, thin vertical stripes, checkerboard and horizontal stripes, respectively. On the other hand, in the Supplementary Information, Appendix B, page iv, the Treatments 1, 2, 3 and 4 are listed as broad vertical stripes, thin vertical stripes, horizontal stripes and checkerboard, respectively.

Reference

Barron and MV Srinivasan (2006) Visual regulation of ground speed and headwind compensation in freely flying honey bees (*Apis mellifera* L.). *The Journal of Experimental Biology* 209, 978-984.

Reviewer #2 (Remarks to the Author):

The authors have addressed all of my concerns and comments, and have greatly improved the clarity of their manuscript. However, I still have one major concern with the interpretation of their

swim speed results, and a few minor comments I would suggest to address before publication.

Major concern: I would like to propose an alternative interpretation of the swim speed results. The authors conclude that the mechanism underlying swimming speed is likely not the same as the one underlying odometry, because the change in swim speed upon changes in spatial frequency are not of the same magnitude. Thus, they propose the underlying mechanism is "largely insensitive to variations in spatial frequency, but swimming speed control was compromised upon removing translational optic flow information entirely. The visual control of swimming speed by *Rhinecanthus* therefore appears to share more similarities with a mechanism measuring the angular rate of visual features used by flying and walking terrestrial animals" (229-233).

When looking at the updated figure, and the statistical results, then there is a highly significant difference in swim speed for the two spatial frequencies (Tukey pairwise comparisons across treatments - Test 1: Test 2, $z = 6.16$, $p < 0.001$; Test 2: Test 3, $z = 2.71$, $p = 0.0340$ -- sidenote: the results from Test 2:Test 4 seem to be missing). Even if absolute magnitude of the change in swim speed is small, the statistical results suggest it is very consistent, and thus I do not see any grounds to dismiss this finding.

Also note: the change in swim speed upon halving the spatial frequency is similar in magnitude to the changes upon removing translational optic flow cues entirely. Which is also (approx.) the case for the distance estimate results. Thus, in relative terms, the same differences between doubling spatial frequency and removing translational optic flow cues in swim speed were observed, as in distance estimated.

But why was the absolute magnitude of change so much lower? Might it be that the fish use a multisensory mechanism to regulate swim speed? Which includes information from the lateral line, which sensed the laminar flow of water in the tunnel, with visual feedback. Depending on the relative strengths of the inputs, vision or water flow sensation gain the upper hand over speed control. In the given setup, vision might only have had a weak contribution, and thus only modulated speed control slightly.

Thus, in my understanding, the speed results are entirely consistent with the hypothesis that speed is estimated by the same visual pathway as distance travelled, which is sensitive to spatial frequency. The speed estimate is then integrated with the sensory information from the lateral line to set the swim speed of the animal. This interpretation explains the statistically significant differences between the tests, and the smaller magnitude of difference. The authors interpretation that speed control is "largely insensitive to variations in spatial frequency" is not consistent with their statistical results.

Minor comments:

Some figure legends still report results in present tense.

I could not find the number of trials that each fish was tested in for the different conditions. These are also not obvious from the figures, as individual datapoints are not shown. It might be an oversight on my part, but therefore worth to point out where this information can be found in the figure legends or methods section.

I. 156 "movement control" could include position control such as the optomotor response, this needs to be phrased more precisely as optomotor based movement control is not independent of spatial frequency

L175-177 I would suggest to provide an explanation for why a reduction in speed by 18% is not biologically relevant.

Reviewer #3 (Remarks to the Author):

The authors have made significant changes to the manuscript to address the reviewers' comments. I believe that the comments (both mine and those of the other reviewers) have been addressed and that the clarity of the manuscript has been improved.

Reviewer #1 (Remarks to the Author):

Comment:

I thank the authors for addressing most of the comments in my earlier review.

However, I still find it difficult to be convinced from the data that optic flow plays a major role in regulating the swimming speed. The speed increases by a very small fraction (less than 10%, Fig. 4) when the tunnel is lined with axial stripes (which should almost completely eliminate the optic flow cues). This suggests that the role of optic flow in regulating swimming speed is, at most, very minor. In honeybees, on the other hand, the flight speed increases by a factor of about 3 when the checkerboard patterns lining the tunnel wall are replaced by axial stripes [see, for example, Fig. 2, Barron and Srinivasan (2006)]. To my mind, the results of the experiments investigating the control of swimming speed strongly suggest that the major regulator of swimming speed is a non-visual cue, possibly hydrodynamic, as I had suggested in my earlier review, and cited a number of studies that have already provided evidence for this. I realize that investigating the role of hydrodynamic cues would be a new effort that is probably beyond the scope of the present study, but I think that it would be important to at least discuss this possibility clearly in the Discussion, because that is what the data seems to suggest. The authors now raise this possibility briefly in the revised Discussion and cite a new reference (Scholtyssel et al., 2014), but dismiss it immediately on the grounds of species differences. The comprehensive review article on the control of fish swimming (Liao 2007) that I had alluded to in my earlier review, which the authors cite in the first version of the manuscript, but not in the second version, clearly states:

“For the majority of fishes, the two most important sensory modalities for swimming are vision and the lateral line sense. The lateral line system detects flow velocity and acceleration via a series of mechanosensory hair cells that are distributed on or just under the skin along the head and body (Coombs et al. 1989). Studies have shown that fishes use both hydrodynamic cues (Dijkgraaf 1963; Montgomery et al. 1997; Engelmann et al. 2000; Coombs et al. 2001) and high-contrast visual cues (Ingle 1971) to swim steadily in uniform flow or in still water. There is a degree of functional redundancy between these two modalities such that blocking the lateral line does not alter the ability to swim in uniform flow if vision is kept intact (Dijkgraaf 1963). In a more complex hydrodynamic environment devoid of visual cues, blocking the lateral line interferes with the ability to swim in steady and unsteady flows. Specifically, fish swimming in the dark without a functional lateral line (i.e. lacking both superficial and canal neuromasts) entrain behind a cylinder less than fish with an intact lateral line (Montgomery et al. 2003).”

Thus, there is a wealth of published evidence to indicate that the lateral line organs play an important role in the control of swimming. It seems to me that the present study would convey a more truthful and meaningful message to the reader if it concluded that optic flow has virtually no influence on swimming speed (which must therefore be regulated by other cues).

Response:

We thank reviewer 1 for these comments, and have updated our manuscript accordingly to stress that the changes in swimming speed with visual treatment are small compared to the effects on distance estimates. We are hesitant to conclude that swimming speeds are not visually controlled at all because changes in swimming speed are observed in response to visual manipulations. These changes also mirror the trends seen in the distance estimation data, even if the magnitude of the changes differ. There is also a consistent increase in swimming speed variability when optic flow is removed, exhibited by all tested individuals. We therefore follow by suggesting swimming speeds are measured as a result of the integration of multiple sensory inputs, with visual information playing a more minor role compared to non-visual inputs, such as hydrodynamic inputs to the lateral line. We have retained the ecological discussion contrasting zebrafish with *Rhinecanthus* because although our results indicate that *Rhinecanthus* only exhibits minor swimming speed responses to optic flow manipulations, this still contrasts the complete lack of response to optic flow manipulations as exhibited by zebrafish in the Scholtyssek (2014) study. Updated sections can be found in lines 140-149, 173-183, and 228-244.

Comment: There seems to be an inconsistency in the numbering of the tests. In the main text, Tests 1, 2, 3 and 4 pertain to the broad vertical stripes, thin vertical stripes, checkerboard and horizontal stripes, respectively. On the other hand, in the Supplementary Information, Appendix B, page iv, the Treatments 1, 2, 3 and 4 are listed as broad vertical stripes, thin vertical stripes, horizontal stripes and checkerboard, respectively.

Response: This was an error on my part in the Supplementary Information – I have corrected this to list Treatments 1, 2, 3, and 4 as broad vertical stripes, thin vertical stripes, checkerboard and horizontal stripes, as they are listed in the main manuscript.

Reference

Barron and MV Srinivasan (2006) Visual regulation of ground speed and headwind compensation in freely flying honey bees (Apis mellifera L.). The Journal of Experimental Biology 209, 978-984.

Response: We have included this reference in the main text to compare our fish with the response of honeybees.

Reviewer #2 (Remarks to the Author):

The authors have addressed all of my concerns and comments, and have greatly improved the clarity of their manuscript. However, I still have one major concern with the interpretation of their swim speed results, and a few minor comments I would suggest to address before publication.

*Major concern: I would like to propose an alternative interpretation of the swim speed results. The authors conclude that the mechanism underlying swimming speed is likely not the same as the one underlying odometry, because the change in swim speed upon changes in spatial frequency are not of the same magnitude. Thus, they propose the underlying mechanism is “largely insensitive to variations in spatial frequency, but swimming speed control was compromised upon removing translational optic flow information entirely. The visual control of swimming speed by *Rhinecanthus* therefore appears to share more similarities with a mechanism measuring the angular rate of visual features used by flying and walking terrestrial animals” (229-233).*

When looking at the updated figure, and the statistical results, then there is a highly significant difference in swim speed for the two spatial frequencies (Tukey pairwise comparisons across treatments - Test 1: Test 2, $z = 6.16$, $p < 0.001$; Test 2: Test 3, $z = 2.71$, $p = 0.0340$ -- sidenote: the results from Test 2:Test 4 seem to be missing). Even if absolute magnitude of the change in swim speed is small, the statistical results suggest it is very consistent, and thus I do not see any grounds to dismiss this finding.

Also note: the change in swim speed upon halving the spatial frequency is similar in magnitude to the changes upon removing translational optic flow cues entirely. Which is also (approx.) the case for the distance estimate results. Thus, in relative terms, the same differences between doubling spatial frequency and removing translational optic flow cues in swim speed were observed, as in distance estimated.

But why was the absolute magnitude of change so much lower? Might it be that the fish use a multisensory mechanism to regulate swim speed? Which includes information from the lateral line, which sensed the laminar flow of water in the tunnel, with visual feedback. Depending on the relative strengths of the inputs, vision or water flow sensation gain the upper hand over speed control. In the given setup, vision might only have had a weak contribution, and thus only modulated speed control slightly.

Thus, in my understanding, the speed results are entirely consistent with the hypothesis that speed is estimated by the same visual pathway as distance travelled, which is sensitive to spatial frequency. The speed estimate is then integrated with the sensory information from the lateral line to set the swim speed of the animal. This interpretation explains the statistically significant differences between the tests, and the smaller magnitude of difference. The authors interpretation that speed control is “largely insensitive to variations in spatial frequency” is not consistent with their statistical results.

Response: We thank the reviewer for this input, and concede that this is a better interpretation of our results. We have updated our manuscript accordingly to stress that although the changes in swimming speed according to visual information changes are small, they are consistent with the results observed for the distance estimates. This suggests that the control of swimming speed and odometry are underpinned by the same motion-sensitive mechanism, but that swimming speeds are predominantly controlled by alternative non-visual mechanisms, such as hydrodynamic inputs to the lateral line. Updated sections can be found in lines 140-149, 173-183, and 228-244.

Minor concerns:

Comment: Some figure legends still report results in present tense.

Response: Figure legends have been updated to the past tense.

Comment: I could not find the number of trials that each fish was tested in for the different conditions. These are also not obvious from the figures, as individual datapoints are not shown. It might be an oversight on my part, but therefore worth to point out where this information can be found in the figure legends or methods section.

Response: We have included trial numbers by fish in the figure legends, and in the supplementary information. We also include a data management document to refer to testing sessions and trial numbers in detail.

l. 156 “movement control” could include position control such as the optomotor response, this needs to be phrased more precisely as optomotor based movement control is not independent of spatial frequency

Response: We have replaced this with: ‘When using optic flow for odometry, movement speeds, and centering responses through gaps and tunnels, terrestrial animals measure the angular motion of features across the retina, a mechanism that is independent of the spatial frequency of the environment.’

L175-177 I would suggest to provide an explanation for why a reduction in speed by 18% is not biologically relevant.

Response: This comment has been removed in line with the new interpretation of the swimming speed results.

Reviewer #3 (Remarks to the Author):

The authors have made significant changes to the manuscript to address the reviewers' comments. I believe that the comments (both mine and those of the other reviewers) have been addressed and that the clarity of the manuscript has been improved.

REVIEWERS' COMMENTS:

Reviewer #1 (Remarks to the Author):

I thank the authors for adequately addressing the main remaining issue that I had raised.

Reviewer #2 (Remarks to the Author):

The authors have significantly improved the clarity and consistency of the manuscript with their last revision, and addressed all of my comments. I therefore endorse publishing the study in the current form.

Reviewer #3 (Remarks to the Author):

I think that the authors have addressed the other two reviewer's concerns about the speed interpretation adequately. I just have one concern about their final sentence on line 261 as I do not believe that they have at all revealed 'a relationship between an animal's habitat and movement ecology' in this paper.